# Motivators and barriers to engagement with evidence-based practice among medical and dental trainees from the UK and Republic of Ireland: a national survey

Bosun Hong,[1,2,3] Eoin Daniel O'Sullivan [2,4] Christin Henein,[2,5,6,7] Christopher Mark Jones [2,8]

For numbered affiliations see end of article.

**Correspondence to**
Dr Christopher Mark Jones; c.jones1@leeds.ac.uk

## ABSTRACT

**Objectives** To explore the extent to which doctors and dentists in training within the UK and Republic of Ireland (RoI) engage in and with evidence-based practice (EBP), and to identify motivators and barriers to them doing so.

**Design** An observational, prepiloted web-based survey developed by a trainee-led focus group.

**Setting** The survey instrument was disseminated to doctors and dentists in training within the UK and RoI during June 2017 via social media and through deaneries, Royal Colleges and specialty-specific mailing lists.

**Participants** Data from 243 trainees were analysed; 188 doctors from 31 specialties and 55 dentists from 9 specialties. Responses were received from trainees at all stages of postgraduate training though the overall response rate was low.

**Primary and secondary outcome measures** The motivators and barriers to, and the extent of, trainee engagement with EBP.

**Results** Cronbach's α was 0.83. Most trainees (87.6% (n=148) of doctors and 75.1% (n=39) of dentists) consulted the evidence base at least monthly, while 23.1% [n=39 doctors, 12 dentists] of both specialties did so daily. The two most commonly cited barriers to engagement with EBP for both doctors and dentists, respectively, were insufficient time (57.6% (n=95) and 45.1% (n=23)) and a tendency to follow departmental practice (40.6% (n=67) and 45.1% (n=23)). Key motivators for EBP included curiosity, following the example set by senior colleagues and a desire to avoid harm. Most trainees reported high levels of confidence interpreting evidence yet for 26.8% (n=45) of doctors and 36.5% (n=19) of dentists, medical hierarchy would impede them querying a colleague's management plan based on their own reading of the evidence.

**Conclusions** Time, accepted departmental practice and the behaviour of senior clinicians all highly impact on trainee engagement with EBP. Given the low response rate, the extent to which these data represent the overall population is unclear.

## Strengths and limitations of this study

► National survey detailing medical and dental trainees' level of engagement with evidence-based practice.

► Provides policymakers, trainers and healthcare organisations with evidence for areas in which trainees' engagement with evidence-based practice can be encouraged and facilitated.

► Reach extends across the UK and Republic of Ireland but results may not represent trainees' experiences in other countries.

► Both dentists and doctors in training sampled but modest sample size may limit generalisability.

## INTRODUCTION

Evidence-based practice (EBP) describes the integration of a patient's values and a clinician's expertise with the contemporary scientific evidence base.[1] The concept encourages the implementation of knowledge gained through systematic scientific discovery into clinical decisions along with psychological rationale, individual clinical experience and patient preference. In doing so, it reduces the potential impact of physician bias while empowering all clinicians, including those with less experience such as trainees, to make decisions informed by high-quality evidence.[2] There is significant evidence for the overall benefit of implementing EBP.[3–5]

Within the UK and the Republic of Ireland (RoI), EBP is firmly embedded within both medical and dental curricula. Reflecting this, the General Medical Council recommends that new doctors must be able to 'access and analyse reliable sources of current clinical evidence and guidance and have established methods for making sure their practice is consistent with these'.[6] Similarly, the General

Dental Council stipulates that on registration, dental trainees should be able to apply an evidence-based approach and use critical thinking skills.[7]

Despite these recommendations and the known tangible benefits of EBP to patient outcomes and care, it is recognised that clinical practice is not universally 'evidence based'.[3 4 8] The barriers to EBP have been studied previously and are numerous.[9–11] Commonly cited examples include inadequate knowledge or awareness of relevant evidence, a lack of individual motivation to engage with the evidence base and difficulty reconciling available evidence with a clinical question.[9]

The extent to which these barriers specifically apply to postgraduate medical and dental trainees, and the degree to which they therefore engage with EBP, is for the most part unknown. Much of the evidence cited to date is instead drawn from grouped populations of undergraduate and postgraduate trainees, or from cohorts consisting of both trainees and more senior clinicians who are beyond completion of training.[10 12] The characteristics of the working environment, clinical responsibilities and drivers for learning of postgraduate trainees may differ from those of non-training clinicians or undergraduates. Hence, an understanding of the role the postgraduate training environment plays in moulding trainees' desire and ability to engage with the evidence base is important. It may, for instance, guide regulators and educators alike in designing interventions to promote the translation of EBP teaching into clinical work for this population. It may also provide training programme coordinators and senior clinicians with an insight into how best to support their more junior colleagues to engage in EBP.

The Cochrane UK & Ireland Trainees Advisory Group was established as the trainee-led arm of Cochrane UK & Ireland in 2016. It is formed of 15 medical and dental trainees from across the UK and RoI who represent a diverse range of specialties and who are together tasked with enhancing trainee engagement with EBP. Given the lack of evidence concerning trainees' engagement with EBP in the contemporary UK and RoI healthcare systems, the Trainees Advisory Group undertook a national survey focused on trainees' use of the evidence base in routine clinical practice and the wider motivators for and barriers to use of the evidence base among these doctors and dentists in training.

## METHODS
### Study design
We conducted a cross-sectional study targeting doctors and dentists in postgraduate training from across the UK and RoI using an anonymous electronic survey between June and October 2017.

### Survey instrument
Members of the Cochrane UK & Ireland Trainees Advisory Group formed a focus group in order to develop an online survey specifically for this study. The rationale for designing a bespoke instrument for this survey was based on our literature review which revealed no validated instrument to cover all areas of focus for our survey. Drawing on their own experiences, the Trainees Advisory Group defined the following as areas of focus for the survey: (A) trainee engagement with EBP, (B) motivators and barriers to engagement with EBP; and (C) trainees' preferences towards approaches to enhance engagement in EBP. Engagement in this context was defined as maintaining sufficient knowledge of the contemporary evidence base relevant to one's clinical role to influence views of or specific practice within that role.

The survey instrument was devised and refined through an iterative process that included two pilot surveys with 19 postgraduate trainees who represented 13 specialties in the fields of medicine and dentistry, and seven geographical locations within the UK and RoI. In its final form (online supplementary file: web-based survey instrument) the survey instrument consisted of 14 closed-ended questions, five of which also permitted optional free-text response and two of which incorporated a 5-point Likert scale. Included questions focused on the aforementioned areas of interest defined by the focus group. Demographic data were also collected and included each trainee's specialty and training stage, prior academic training and their location, subdivided as either RoI or by UK deanery. We collected data using a cloud-based online survey program (SurveyMonkey, California, USA).

### Study population and survey dissemination
Trainees in a recognised UK or RoI training post within any medical or dental specialty formed the target population for this study. Full registration with the General Medical Council was not required in order to ensure that responses from Foundation Year 1 doctors were considered. UK and RoI trainees are widely geographically distributed and are overseen by a large number of professional organisations, including regional deaneries and various Royal Colleges. There is no single point of contact for either doctors or dentists in training. In order to enhance coverage of trainees eligible to participate in this survey, we used a phased approach to its dissemination that over a 3-week period incorporated both a social media arm and direct approaches to trainees via email and online training portfolios (see figure 1).

In week 1 (commencing 30 June 2017), we disseminated the survey via email to relevant professional organisations with oversight of doctors and dentists in training across the UK and RoI. We also used the email marketing service MailChimp to disseminate the survey to medical deaneries across the UK and to 49 trainees subscribed to the Cochrane UK & Ireland Trainees Advisory Group newsletter (see online supplementary table 1). MailChimp enabled us to check whether our emails were opened by the recipient organisations but could not facilitate tracking of the actual survey response rate as the number of trainees receiving the survey link, including via other modalities such as social media, was not known.

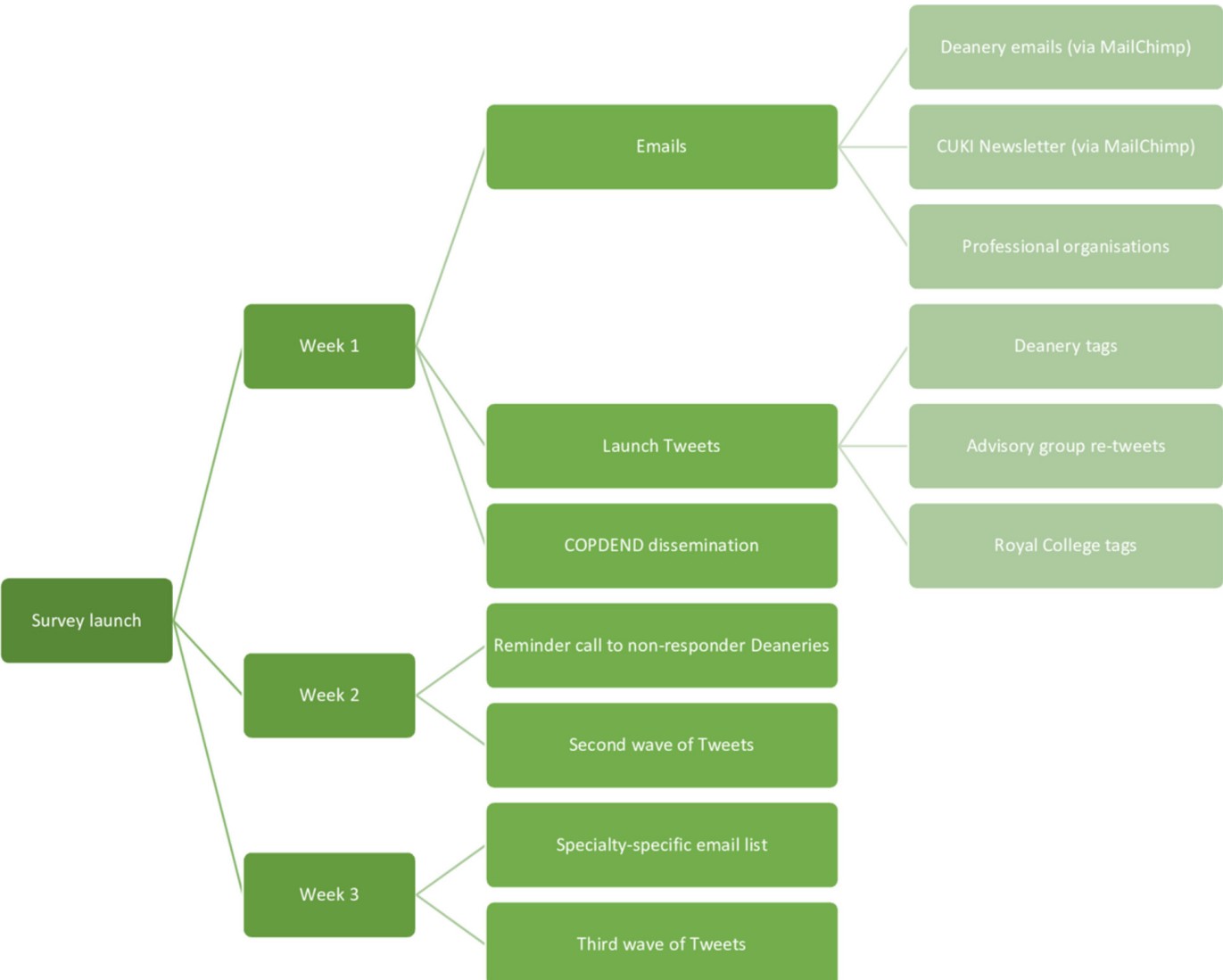

**Figure 1** A flow chart depicting the survey dissemination approach, including via social media. COPDEND, Committee of Postgraduate Dental Deans and Directors; CUKI, Cochrane UK & Ireland.

We also distributed to UK dental deaneries following approval from the UK Committee of Postgraduate Dental Deans and Directors. In both instances, administrators within these professional bodies were asked to distribute the survey to trainees overseen by their organisation via direct email or through use of the National Health Service (NHS) training ePortfolio, to which all medical trainees have access.

During this same week, a link to the survey was posted online using the Cochrane UK Twitter account (@ CochraneUK), which at the time had in excess of 40 000 followers. In line with an a priori planned social media strategy, this link was retweeted nine times by this same account and by members of the Cochrane UK & Ireland Trainees Advisory Group within the same week. Tweets were flagged for the attention of the Twitter accounts of relevant Royal Colleges and other bodies overseeing trainees.

In week 2, we once again contacted those professional bodies who had been contacted during week 1 but who had not responded to the first survey approach. Contemporaneously, further tweets outlining the survey were distributed by the Cochrane UK & Ireland Twitter account. These continued into the third week. The survey subsequently remained open until October 2017 in order to permit further responses to be collected.

### Data analysis

The unit of analysis of this study is the individual respondent (trainee). Data are provided for doctors and dentists separately where possible. There were insufficient participant numbers to analyse by specialty. Data are shown for separate stages of training where numbers allow. Descriptive analyses were undertaken using a statistical software R (V.3.4.3, R Core Team, 2013). We computed the frequency of answers and where relevant expressed this as a percentage of total respondents. For Likert-type scaled response answers, we assumed answers to be an interval-level measurement to allow visualisation using bar plots.

We conducted a thematic analysis of the free-text responses using an inductive approach, following the phases of thematic analysis outlined by Clarke *et al*.[13] Having become familiarised with the data, we identified initial codes using a qualitative data analysis software (NVivo V.12, QSR International, 2018) and organised the codes into potential themes. The identified themes were reviewed and refined before final categorisation. In addition, we have searched for and retained deviant responses that do not fit into the themes but nevertheless make a valuable contribution to the overall message of this work.

### Patient and public involvement

There was no direct patient involvement in the design or conduct of this study.

## RESULTS

### Internal consistency

The survey had high internal consistency with a Cronbach's α of 0.83.

### Respondent demographics

A total of 243 responses were received; 188 (83.3%) from doctors in 31 specialties, and 55 (16.7%) from dentists in 9 specialties (see table 1). Most (53% (n=123)) trainees were specialty training 3 level or above in seniority, and the majority (70% (n=166)) had not previously undertaken formal postgraduate academic training. Responses were received from trainees in RoI and in 20 out of 21 UK deaneries. The geographical distribution of respondents is summarised in table 2.

### Engagement with EBP

To assess trainees' level of engagement with EBP, those responding to the survey were asked how frequently they would refer to published literature to determine the evidence base for a specific action or intervention (figure 2A). The majority of doctors (87.6% (n=148)) and dentists (75.1% (n=39)) reported consulting the evidence base at least monthly. An identical proportion (23.1% (n=39 doctors and 12 dentists)) of the two professions consulted the evidence base daily.

Trainees were additionally asked to report which sources of evidence they had previously consulted in order to inform their practice (figure 2B). For both doctors and dentists, national (95.8% (n=160) and 88.5% (n=46), respectively) and local (81.4% (n=136) and 67.3% (n=35)) guidance, National Institute for Health and Care Excellence (NICE) Clinical Knowledge Summaries (82.6% (n=138) and 86.5% (n=45)), systematic reviews (85.6% (n=143) and 84.6% (n=44)) and published original research (79.6% (n=133) and 63.5% (n=33)) formed the principal sources of information.

Trainees' self-ratings of their confidence in searching evidence and interpreting basic statistics were strongly negatively skewed (figure 3A,B), though this distribution became normal when trainees with prior academic

| Table 1 | Survey respondents by profession and specialty | | | |
|---|---|---|---|---|
| | Doctors (n=188) | | Dentists (n=55) | |
| | n | % | n | % |
| **Specialty** | | | | |
| Acute internal medicine | 1 | 0.5 | | |
| Anaesthetics | 7 | 3.7 | | |
| Cardiology | 1 | 0.5 | | |
| Chemical pathology | 1 | 0.5 | | |
| Clinical radiology | 2 | 1.0 | | |
| Community sexual and reproductive health | 12 | 6.4 | | |
| Dermatology | 2 | 1.1 | | |
| Emergency medicine | 5 | 2.7 | | |
| Foundation training | 24 | 12.8 | 27 | 49.1 |
| General medicine | 5 | 2.7 | | |
| General practice | 6 | 3.2 | 1 | 1.8 |
| General psychiatry | 2 | 1.1 | | |
| General surgery | 6 | 3.2 | | |
| Geriatric medicine | 4 | 2.1 | | |
| Haematology | 1 | 0.5 | | |
| Oral histopathology | | | 1 | 1.8 |
| Intensive care medicine | 1 | 0.5 | | |
| Infectious diseases | 2 | 1.1 | | |
| Neurosurgery | 1 | 0.5 | | |
| Obstetrics and gynaecology | 20 | 10.6 | | |
| Occupational medicine | 2 | 1.1 | | |
| Ophthalmology | 2 | 1.1 | | |
| Oral and maxillofacial surgery* | 1 | 0.5 | 3 | 5.5 |
| Oral medicine | | | 8 | 14.5 |
| Oral surgery | | | 5 | 9.1 |
| Orthodontics | | | 2 | 3.6 |
| Paediatric and perinatal pathology | 1 | 0.5 | | |
| Paediatric dentistry | | | 4 | 7.3 |
| Paediatric medicine | 9 | 4.8 | | |
| Palliative medicine | 2 | 1.1 | | |
| Periodontology as a monospecialty | | | 1 | 1.8 |
| Plastic surgery | 1 | 0.5 | | |
| Public health medicine | 57 | 30.3 | | |
| Renal medicine | 3 | 1.6 | | |
| Respiratory medicine | 2 | 1.1 | | |
| Restorative dentistry | | | 2 | 3.6 |
| Special care dentistry | | | 1 | 1.8 |
| Urology | 1 | 0.5 | | |

Continued

**Table 1** Continued

|  | Doctors (n=188) | | Dentists (n=55) | |
|---|---|---|---|---|
|  | n | % | n | % |
| Vascular surgery | 1 | 0.5 |  |  |
| Unknown | 3 | 1.6 |  |  |

*Oral and maxillofacial surgery is recognised as a specialty by the General Medical Council and as such their specialty trainees are registrants of the General Medical Council. In addition to specialty training, the specialty facilitates dental core training and therefore the majority of their junior staff are dentists.

**Table 2** Survey respondents by profession, location of training and training stage

|  | Doctors (n=188) | | Dentists (n=55) | |
|---|---|---|---|---|
|  | n | % | n | % |
| **Respondent's location of training** | | | | |
| Scotland | 13 | 6.9 | 25 | 45.5 |
| East Midlands, England | 9 | 4.8 | 6 | 10.9 |
| East of England | 5 | 2.7 | 3 | 5.5 |
| London, England | 33 | 17.6 | 2 | 3.6 |
| North East, England | 3 | 1.6 | 1 | 1.8 |
| North West, England | 10 | 5.3 | 4 | 7.3 |
| South East, England | 22 | 11.7 | 2 | 3.6 |
| South West, England | 17 | 9.0 | 2 | 3.6 |
| West Midlands, England | 15 | 8.0 | 6 | 10.9 |
| Yorkshire and the Humber, England | 19 | 10.1 | 1 | 1.8 |
| Wales | 4 | 2.1 | 0 | 0.0 |
| Northern Ireland | 3 | 1.6 | 0 | 0.0 |
| Republic of Ireland | 28 | 14.9 | 2 | 3.6 |
| Unknown | 7 | 3.7 | 1 | 1.8 |
| **Training stage** | | | | |
| Foundation years | 22 | 11.7 | 28 | 50.9 |
| Core/specialty trainee 1/2 | 39 | 20.7 | 15 | 27.3 |
| Specialty trainee 3 | 30 | 16.0 | 4 | 7.3 |
| Specialty trainee 4 | 31 | 16.4 | 2 | 3.6 |
| Specialty trainee 5 | 20 | 10.6 | 2 | 3.6 |
| Specialty trainee 6 | 9 | 4.8 | 1 | 1.8 |
| Specialty trainee 7 | 8 | 4.3 | 0 | 0.0 |
| Specialty trainee 8 | 0 | 0.0 | 2 | 3.6 |
| RoI—intern | 3 | 1.6 | 0 | 0.0 |
| RoI—senior house officer | 4 | 2.1 | 1 | 1.8 |
| RoI—specialty registrar | 10 | 5.3 | 0 | 0.0 |
| Unknown | 12 | 6.4 | 0 | 0.0 |

RoI, Republic of Ireland.

experience were removed from the analysis. As highlighted by figure 3, confidence in these key skills also increases with training stage.

When asked for their perception of social media and the internet for education to support their EBP, the majority of respondents (72.0% (n=157)) were supportive of its use. Popular forms of engagement were via email updates (53.7% (n=117)), an online journal club (47.3% (n=101)) and through a podcast (40.4% (n=88)).

### Motivators for EBP

There was some variation between doctors and dentists in the factor respondents highlighted as their motivation for consulting the evidence base (figure 4A). For dentists, the most frequently cited reason was to better understand how clinical decisions are made (84.3% (n=43)), a curiosity noted by 54.6% (n=90) of responding doctors. Conversely, while 42.4% (n=123) of doctors reported that consulting the evidence base was routine practice in their specialty, this was the case for just 17.6% (n=9) of dentists. For both doctors and dentists, lack of a relevant guideline (37.0% (n=61) and 39.2% (n=20), respectively), encouragement from senior colleagues (37.0% (n=61) and 45.1% (n=23)) and an awareness of the potential for a negative clinical outcome from not consulting the evidence base (34.6% (n=57) and 41.2% (n=21)) were other frequently reported motivators for engagement in EBP.

These findings were supported by thematic analysis of free-text comments, which revealed that motivators for EBP included trainees' desire to provide a good standard of care at either the individual patient level, or for public health trainees, at the population level. The responses captured the inspiration to do so, which included examples of harm caused to patients by non-EBP, involvement in research projects and following the example set by senior colleagues:

> I think it is important to review literature as often ingrained practice has no real proof of benefit. I also think that by understanding the evidence you learn better and you understand its applicability and are better able to decide if it applicable for your patient cohort.

> The ethical implications of continuing to practice in an evidence-light environment following the withdrawal of the Liverpool Care Pathway

> …My consultants are also very much into [EBP] & quote papers all the time. I feel that I'm expected to know the evidence behind the treatments I offer patients & I think this is a good thing as it makes me much more confident at recommending a particular treatment to a patient.

### Barriers to engagement with EBP
#### Time, access and local practices

Barriers to EBP reported by trainees are summarised in figure 4B. The two most commonly cited barriers for

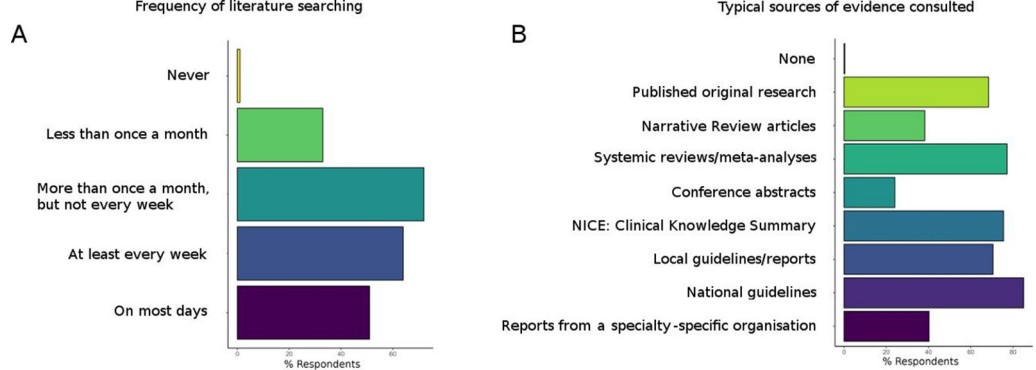

**Figure 2** Trainee engagement with evidence-based practice, including (A) the frequency with which trainees consult published literature in order to evaluate a clinical problem and (B) the sources trainees would typically consult in order to assess the evidence base. NICE, National Institute for Health and Care Excellence.

doctors and dentists, respectively, were insufficient time (57.6% (n=95) and 45.1% (n=23)) and a tendency to follow departmental practice (40.6% (n=67) and 45.1% (n=23)). Limited access to relevant evidence from the workplace was also commonly reported as a barrier, though more frequently among doctors (40.6% (n=67)) than dentists (25.5% (n=13)).

Thematic analysis of free-text responses supported these data, and highlighted that trainees considered time and access as important barriers to EBP:

> By far the greatest barrier to taking an evidence-based approach to my work is the lack of access to the evidence. I am a public health registrar with a passion for the wider determinants of health. At the moment I have very limited Athens access through the training scheme, but following this I'm not sure if I will even have that…

> I am limited by the insufficient electronic journal access provided as part of NHS Athens, which is inferior to the access provided by a good university account. I do manage to work around this, but it takes extra time and limits the sources I can use.

Some trainees felt that the access issue was twofold: electronic access to articles being the first hurdle, and having the information technology facility within the NHS premises to allow access to the information being the next one:

> …the evidence base is spread across multiple resources, some of which are, with the best will in the world difficult to use in day to day practice, from crappy, locked down NHS computers. If they are free to use at all.

> Poor wifi signal often frustrates me when I want to look at a resource such as NICE or UptoDate on the wards.

Some trainees felt that the use of evidence was not prioritised in postgraduate training despite its importance, and that those who require further training on this may struggle to find the time due to other priorities:

> We are not taught how to use the evidence base. Yeah sure, the local library might offer some introductory courses (if you are in a good hospital) but it is unacceptable that 'moving & handling' is a mandatory

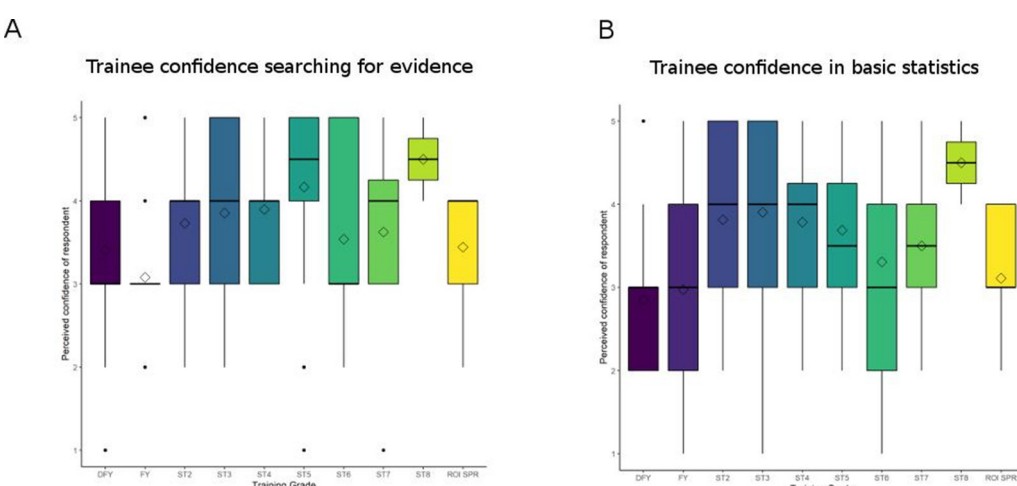

**Figure 3** Perceived confidence in searching evidence (A) and interpreting basic statistics (B) on a scale of 1 (least confident) to 5 (most confident).

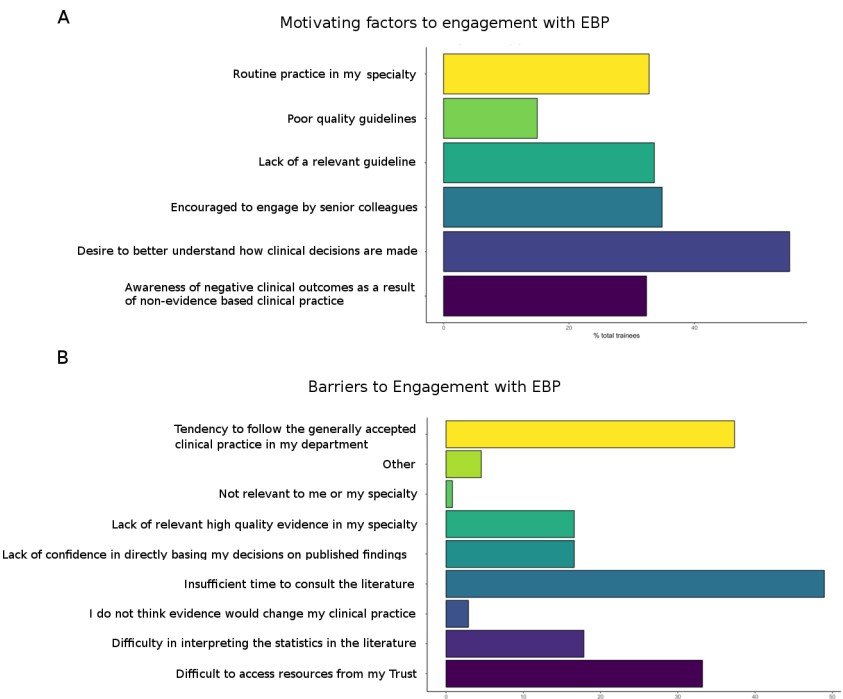

**Figure 4** A summary of reported (A) motivators and (B) barriers to engagement with evidence-based practice (EBP).

course but searching and using the evidence base is treated as optional.

I was lucky enough to learn about evidence-based research at medical school and taught how to conduct literature reviews but it is something I just don't have the time for now.

### Limited evidence

Dentists in training more commonly cited a lack of relevant or high-quality evidence as a barrier than doctors (37.3% (n=19) vs 12.7% (n=21)). Thematic analysis also revealed the challenges trainees face in specialties in which evidence is scarce and where studies have unclearly reported methodology or questionable generalisability to their patients:

…there is a really lack of randomised controlled trials in my specialty and rejection of them by some of the large research groups. The question of how much evidence is enough evidence in palliative medicine remains to be answered.

Reliance on secondary evidence such as local, specialty-specific or national guidelines was evident in the free-text responses. Trainees perceived secondary evidence to be of high quality and therefore their default first choice of avenue for evidence:

I use individual papers very rarely. Think Royal College of Obstetrics and Gynaecology does a good job of synthesising current information and producing guidance on common topics.

Commonly used guidelines have been memorised—for example, asthma, transient loss of

consciousness.

### Hierarchy

When trainees were asked if they would feel comfortable querying a colleague's management plan based on the trainee's reading of the evidence, 57.7% (n=97) of doctors and 51.9% (n=27) of dentists were unconditionally so. Other trainees reported that they would feel confident only if the colleague was more junior or less experienced than them (19.6% (n=33) and 30.8% (n=16)), or that this depended on the situation (15.5% (n=26) and 11.5% (n=6)). A smaller proportion of respondents stated that they would feel uncomfortable about querying a colleague's management in any case (7.1% (n=12) and 5.8% (n=3)).

On analysing the free-text responses, a passive attitude of following generally accepted departmental practice was a frequent feature. It was evident that the working relationship with, or the personality of a senior colleague was more important to trainees than the seniority itself when considering whether to question a senior colleague's decision. Embedded within the responses was a situational analysis process that trainees go through in their mind before deciding to approach their senior colleague. Some trainees have developed a strategy where their decision to question a senior colleague would be conditional—for example, only if the impact to the patient is large, only if they predict a positive response from the colleague and only if they have direct responsibility for care (ie, 'asked to complete the treatment'):

Some colleague's personalities may colour my comfort level to query/discuss management!

Consultants who are confident and progressive would be willing to discuss. Some seniors would find this threatening as they are insecure in their own practice.

…you have to read the situation…

It really depends on so many factors (eg, who is the manager, who are you, how long you have been there, what type of relation you have…) In general, I would only do it if I feel that there will be a positive response. If I feel that it is only going to generate conflict, then, what's the point?

… I pick my fights as such and only raise it if I think it is important/going to make a difference etc.

### EBM malaise

There was also a response which pointed out that a clinician's unfounded preconceived notions may act as a barrier to a scientific enquiry:

The problem is that the concept of 'evidence' has lost its meaning. People use for all sorts of things…I mean that people, when faced by a public health issue, they don't ask themselves 'What does the evidence say about how to resolve this issue?' What usually happens is that they have a pre-defined position and they ask themselves 'What is the evidence that supports my position?' That influences how searches are done, how bias is assessed and so on.

### DISCUSSION

The extent to which doctors and dentists in contemporary training schemes engage with the evidence base, and the barriers and motivators to them doing so, is underexplored. We outline here a nationwide survey of UK and RoI doctors and dentists from a large number of specialties and at all stages of training. Most actively engage with literature at least monthly, while a quarter does so daily. Dentists in training most frequently cited curiosity as their main motivator for EBP and reported a lack of relevant evidence as a key barrier to further engagement with the evidence base. On the other hand, doctors frequently cited their engagement with EBP in the context of accepted practice for the departments in which they work. For both doctors and dentists, a lack of time and poor access to relevant literature were key barriers to further engagement in EBP. Trainees nevertheless reported high levels of confidence in searching for evidence and interpreting basic statistics. However, the majority also reported that they would not query a more senior colleague based on their own reading of the evidence, thereby potentially limiting the extent to which their confidence translates in to EBP.

The notion that trainees appreciate EBP but do not consider their practice to be consistently evidence based is supported elsewhere in the literature.[14–16] For instance, a survey of pre-registration house officers (currently termed foundation trainees in the UK) in the Mersey Deanery revealed that trainees considered critical appraisal skills to be relevant to their practice yet only just over half felt that their clinical practice was based on the best available evidence.[14] Similarly, a survey of obstetrics and gynaecology trainees in West Midlands revealed that while most trainees perceived EBP as a positive concept, relatively few consistently referred to literature.[15] It is of interest that many of the barriers identified in these more limited specialty-specific studies are reflected in the broad trainee population reported here.

Our findings have a number of additional implications, which the Cochrane UK & Ireland Trainees Advisory Group have reviewed with a view to formulating strategic, concrete plan to promote EBP. The higher reported confidence in EBP of trainees who have undertaken formal postgraduate academic training demonstrates the value of academic training schemes to healthcare settings. Notably, that the hierarchy and departmental practice so considerably shape the degree to which trainees engage in EBP demonstrates the parallel importance of ensuring an appropriate training environment. In view of this, the Trainees Advisory Group plans to collect examples of good practice as a result of postgraduate academic training and positive influence from colleagues or department practice, and disseminate them through its online platforms. The data depicted in our survey additionally demonstrate a clear need to support busy trainees in digesting evidence, whether through directly managing their workload or by ensuring that sources of evidence are readily available and their conclusions efficiently consumable. Other implications of this work are likely to be harder to address. The paucity of relevant evidence highlighted by dental trainees is one such area but may be challenged by promoting greater awareness of EBP within the profession.

### Limitations

The number of trainees reached by the survey is not known. However, there are over 60 000 doctors and nearly 3000 dentists in training in the UK alone.[14 15] The overall response rate was therefore low and the data are prone to responder bias. Responses were however captured from a large number of specialties and good geographical coverage was achieved. The degree to which the data depicted here represent the overall population is, therefore, unclear and the outcomes of this survey require validation in a larger cohort. Further, a number of specialties were not represented or were but by only a small number of respondents, and other specialties such as public health were relatively over-represented. As such, it is impossible to characterise differences in the data between specialties and training stages. The use of email and social media to contact doctors and dentists also provides the potential for selection bias. The extent to which these results are applicable to healthcare settings outside of the UK and RoI is in addition unclear.

This survey nevertheless had a wide geographical reach across both the UK and RoI, incorporated a significant number of specialties and featured responses from both doctors and dentists in training. Internal consistency was also high, as measured by a Cronbach's α of 0.83. The data provide a number of clear themes, some of which are specific to the postgraduate trainee population. For example, the important influence of hierarchy, both as a motivator and a barrier, is a trainee-specific theme that emerged prominently as a result of our focus on the postgraduate trainee population. These themes provide a starting point for further research and intervention from both healthcare professionals and policymakers. The validity of these themes is supported by their concordance with historical evidence,[16–18] and their relevance is reflected from the postmodernising medical careers cohort from which they are derived.

## CONCLUSION

Time and a lack of access to key resources limit the ability of doctors and dentists in training within the UK and RoI to engage with EBP. While the majority of these trainees report high levels of confidence in searching and interpreting statistics in the literature, a significant proportion highlight that hierarchy and departmental practice significantly shape their EBP. The extent to which these data vary between specialties and by stage of training is unknown and requires further research. Healthcare professionals and policymakers should ensure that trainees are provided with the time and ability to efficiently access sources of evidence, and should engender cultures in which EBP is encouraged.

**Author affiliations**
¹Birmingham Dental Hospital, Birmingham, UK
²Cochrane UK & Ireland Trainees Advisory Group, Oxford, United Kingdom
³School of Dentistry, College of Medical & Dental Sciences, University of Birmingham, Birmingham, United Kingdom
⁴Department of Renal Medicine, Royal Infirmary of Edinburgh, Edinburgh, United Kingdom
⁵Institute of Genetic Medicine, Newcastle University, Newcastle, United Kingdom
⁶National Institute for Health Research Biomedical Research Centre, Moorfields Eye Hospital NHS Foundation Trust, London, United Kingdom
⁷Institute of Ophthalmology, University College London, London, United Kingdom
⁸Faculty of Biological Sciences, University of Leeds, Leeds, United Kingdom

**Acknowledgements** We are grateful to trainees for taking the time to respond to this survey and to the various professional bodies who distributed the survey. We are also grateful for support from Cochrane UK, which is funded by the National Institute for Health Research (NIHR), and to Dr Emma Plugge for her critical appraisal of the manuscript.

**Contributors** BH and CMJ devised, piloted and carried out the survey, and collated the data. BH prepared the first draft of the manuscript. BH, EDOS, CH and CMJ led the analysis of data. All authors contributed to the interpretation of results and revisions of the draft manuscript, and have read and approved the final version of the manuscript.

**Funding** BH was supported for the duration of this work by a National Institute for Health Research (NIHR) Academic Clinical Fellowship (ACF) in Oral Surgery. EDOS was supported for the duration of this work by Kidney Research UK. CH was supported for the duration of this work by an NIHR ACF in Ophthalmology. CMJ was supported for the duration of this work by an NIHR ACF in Clinical Oncology and a Wellcome Trust N4 Clinical Research Training Fellowship (203914/Z/16/Z).

**Competing interests** None declared.

**Patient consent for publication** Not required.

**Ethics approval** There was no requirement for specific ethical approval for this study. The survey was distributed to medical and dental trainees across the UK and RoI. Trainees received the survey via an email or ePortfolio link, or may have followed a link to the survey via social media. In all instances neither we nor any other person or professional organisation collected or had access to any identifiable data relating to trainees completing the survey. Indeed, all trainees were reminded at the outset of survey completion that they were under no obligation to complete the questionnaire and that their details and answers would remain anonymous. All participants were informed of the study's purpose and content, and all completed the survey on a device of their choosing to which the researchers had no access and all were free to stop participating at any point. Under NHS Health Research Authority guidelines, NHS Research Ethics Committee (REC) approval is not required for research of this kind in England, Scotland, Wales and Northern Ireland (as per the HRA Decision Toolkit: http://www.hra-decisiontools.org.uk/ethics/). Similarly, under the Health Service Executive guidelines, research of this nature does not require REC approval in the RoI (as per https://www.hse.ie/eng/services/list/5/publichealth/publichealthdepts/research/rec.html). In addition, there was no incentive offered for survey completion.

**Provenance and peer review** Not commissioned; externally peer reviewed.

**Data availability statement** Data are available upon reasonable request.

**ORCID iDs**
Eoin Daniel O'Sullivan http://orcid.org/0000-0002-7709-6595
Christopher Mark Jones http://orcid.org/0000-0002-4513-4964

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
