## [Reviewer comments · BMJ Open]

ARTICLE DETAILS

TITLE (PROVISIONAL)	Motivators & Barriers to Engagement with Evidence Based Practice Amongst Medical & Dental Trainees from the United Kingdom & Republic of Ireland: a National Survey
AUTHORS	Hong, Bosun; O'Sullivan, Eoin; Henein, Christin; Jones, Christopher

VERSION 1 – REVIEW

REVIEWER	Loai Albarqouni Bond University, Australia
REVIEW RETURNED	05-Jul-2019

GENERAL COMMENTS	Thanks for inviting me to review this manuscript. In this article, authors surveyed doctors and dentists working in the United Kingdom and Ireland to identify barriers and facilitators to evidence-based practice. Time, accepted practice, and senior clinicians' behaviour found to be highly influential. Overall, this is well conducted piece of research. There are a few comments that should be taken into consideration. Abstract - Authors should report the findings in terms of absolute numbers as well as percentages (xx, xx%). Also, authors should highlights a major shortcoming of the survey i.e. low response and how this might impact the interpretation of findings. Background - In page 1; lines 7-16: This sentence might be misinterpreted as a call to ignore clinical experience. I would strongly recommend the authors to amend this sentence taking into consideration Haynes et al article https://ebm.bmj.com/content/7/2/36 Methods - Authors should justify why they do not use previously developed relevant surveys?- Would it be possible for authors to track the response rate? Or compare those who participated with national figures?- Authors should report whether they have piloted the questionnaire?- Results - I would suggest reporting the absolute numbers along with the percentages.- Did the author reported any information regarding the age, clinical experience, gender, or the study location? If yes, this need to be included in table 1.- Figure 1 is too small. This should be clear.
--

	- Figure 2-4 need to be reproduced to ensure that they are clear and can be easily interpreted. In its current shape, it is difficult to go through all the data in the figures.
--	--

REVIEWER	Ruiling Guo Idaho State University United States
REVIEW RETURNED	10-Jul-2019

GENERAL COMMENTS	Strengths: This is a fine and interesting study. I can see the researchers have put lots of time and efforts to complete this research. The research methodology used in this study is appropriate. The researchers identified the main motivators and barriers to the EBP engagement among medical and dental trainees. I think the unique part of this study was to target medical and dental trainees specialized in different medical fields from large geographic areas. This study does present some implications for the practice of EBP among healthcare professionals. Weaknesses: 1. The response rate in this study was quite low, given the fact that thousands of potential participants were reached out. However, an online survey usually presents challenges for researchers. As I mentioned, the unique of this study was to reach out to medical and dental trainees from large geographic areas. I hope it reduces some bias. 2. Since this study targets human subjects, the approval from the Institutional Review Board may be needed. The researchers did not mention it in this study. 3. The text in Figures 2 and 4 was not clear enough to be read. Please change the font to ensure the text is readable. 4. Please double check spellings in the manuscript. 5. Some of the references were not completed. Please double check the sources of some references to ensure they are completed. I've really enjoyed reviewing this manuscript. Thank you!
--

VERSION 1 – AUTHOR RESPONSE

Reviewer: 1

Authors should report the findings in terms of absolute numbers as well as percentages (xx, xx%). We have added absolute numbers next to percentage values. Please note that whilst doing so, we spotted a mathematical error in the following sentence of the Results section of the abstract and have amended it as follows:

“Most trainees reported high levels of confidence interpreting evidence yet for 19.6% 26.8% [n=45] of doctors and 30.8% 36.5% [n=19] of dentists, medical hierarchy would impede them querying a colleague’s management plan based on their own reading of the evidence”.

Authors should highlights a major shortcoming of the survey i.e. low response and how this might impact the interpretation of findings.

This is now highlighted in the abstract and is discussed in more detail in the discussion section of the manuscript.

In page 1; lines 7-16: This sentence might be misinterpreted as a call to ignore clinical experience. I would strongly recommend the authors to amend this sentence taking into consideration Haynes et al article <https://ebm.bmj.com/content/7/2/36>

We are grateful to the reviewer for highlighting this and agree. The opening paragraph has been amended in light of this comment and to better reflect Haynes' article.

Authors should justify why they do not use previously developed relevant surveys?

We have added a sentence under 'Survey instrument' in the Methods section to highlight that there was no validated survey instrument available.

Would it be possible for authors to track the response rate? Or compare those who participated with national figures?

We were unfortunately unable to track the response rate, which is now made clear in the 'Study population & survey dissemination' section. A separate comment relating to the response rate in the context of the number of doctors and dentists in training across the sampled region has been added to the limitations section of the manuscript.

Authors should report whether they have piloted the questionnaire?

Please note that this was reported in the second paragraph under 'survey instrument' of the Methods section:

"The survey instrument was devised and refined through an iterative process that included two pilot surveys with 19 postgraduate trainees who represented 13 specialties in the fields of medicine and dentistry, and seven geographical locations within the UK and RoI."

I would suggest reporting the absolute numbers along with the percentages.

Absolute numbers are now reported next to percentage values.

Did the author reported any information regarding the age, clinical experience, gender, or the study location? If yes, this need to be included in table 1.

We have reported on the stage of training of study respondents, rather than age, as depicted in Table 2. Respondent age was not requested as training stage was felt to be a more meaningful measure (i.e. of a participant's experience) against which to evaluate study data. Geographical location of the trainee respondent was captured and presented in Table 2.

Figure 1 is too small. This should be clear. Figure 2-4 need to be reproduced to ensure that they are clear and can be easily interpreted. In its current shape, it is difficult to go through all the data in the figures.

We are grateful to the reviewer for highlighting this and have adjusted these figures accordingly.

Reviewer: 2

The response rate in this study was quite low, given the fact that thousands of potential participants were reached out. However, an online survey usually presents challenges for researchers. As I mentioned, the unique of this study was to reach out to medical and dental trainees from large geographic areas. I hope it reduces some bias.

Thank you. We agree with this comment and in line with additional comments from Reviewer 1 have added further details relating to the response rate to both the 'Limitations' section and to the 'Study population & survey dissemination' section.

Since this study targets human subjects, the approval from the Institutional Review Board may be needed. The researchers did not mention it in this study.

A fuller statement relating to IRB/ethics approval is now included within the Methods section of the manuscript.

1. The text in Figures 2 and 4 was not clear enough to be read. Please change the font to ensure the text is readable.

We have revised all of our figures to make it clearer to the readers

2. Please double check spellings in the manuscript.

This has been done and all spellings are, to our knowledge, accurate.

5. Some of the references were not completed. Please double check the sources of some references to ensure they are completed.

Thank you for spotting this. We have realised that Clark and Braun reference was incomplete and have amended this, as requested.

VERSION 2 – REVIEW

REVIEWER	Loai Albarqouni Australia
REVIEW RETURNED	25-Aug-2019
GENERAL COMMENTS	The authors have successfully revised the manuscript based on reviewers comments.